# From cockroaches to tanks: The same power-mass-speed relation describes both biological and artificial ground-mobile systems

**Alexander Kott[ID]\*, Sean Gart, Jason Pusey**

U.S. Army CCDC Army Research Laboratory, Adelphi, Maryland, United States of America

\* alexander.kott1.civ@mail.mil

**Data Availability Statement:** All relevant data are within the paper and its Supporting information files.

## Abstract

This paper explores whether artificial ground-mobile systems exhibit a consistent regularity of relation among mass, power, and speed, similar to that which exists for biological organisms. To this end, we investigate an empirical allometric formula proposed in the 1980s for estimating the mechanical power expended by an organism of a given mass to move at a given speed, applicable over several orders of magnitude of mass, for a broad range of species, to determine if a comparable regularity applies to a range of vehicles. We show empirically that not only does a similar regularity apply to a wide variety of mobile systems; moreover, the formula is essentially the same, describing organisms and systems ranging from a roach (1 g) to a battle tank (35,000 kg). We also show that for very heavy vehicles (35,000–100,000,000 kg), the formula takes a qualitatively different form. These findings point to a fundamental similarity between biological and artificial locomotion that transcends great differences in morphology, mechanisms, materials, and behaviors. To illustrate the utility of this allometric relation, we investigate the significant extent to which ground robotic systems exhibit a higher cost of transport than either organisms or conventional vehicles, and discuss ways to overcome inefficiencies.

## Introduction

Remarkable regularities are observed with respect to relations among the speed, mass, and energy expenditures of biological organisms. Most of these relations are allometric, i.e., they describe how a function or attribute of organisms changes with their scale (e.g., with mass). Although the accuracy of the relations tends to be limited to an order of magnitude, these relations hold over very large ranges of values and over multiple orders of magnitude. For example, one of the earliest such observations was Kleiber's Law [1], which states that for the vast majority of animals–from tiny mouse to huge elephant–the organism's metabolic rate is approximately proportional to the organism's mass to the 3/4 power; the data for all biological organisms fall on the same curve.

**Funding:** The authors received no specific funding for this work.

**Competing interests:** The authors have declared that no competing interests exist.

In a similar allometric fashion, it has been observed that for all biological organisms, the maximum speed $V$ of animals increases proportionately to their mass $M$ to the power of 1/6, i.e., $V \sim M^{1/6}$, even when the mass of organisms varies by many orders of magnitude [2,3].

Another allometric law refers to the metabolic cost of transport (CoTm), which is the metabolic energy consumption that an organism requires to move a unit of mass of its body over a unit of distance. Equivalently, the amount of metabolic energy per unit time $Pm$ required for the organism to move with speed $V$ per unit of its mass M is CoTm = $Pm/(M \cdot V)$. For all biological organisms, CoTm diminishes approximately with mass to the power of -1/3, i.e., $Pm/(M \cdot V) \sim M^{(-1/3)}$ [4–8]. Metabolic energy is the energy that an organism obtains through its food, though typically measured as the organism's consumption of oxygen [9].

Unlike metabolic energy, the mechanical energy of an organism $P$ is produced by the organism's muscles in order to deliver forces by which it propels itself relative to the terrain and surrounding media, and moves its limbs with respect to the rest of its body. It has been observed that unlike the metabolic CoTm, the mechanical CoT = $P/(M \cdot V)$ remains approximately constant, or at least within the same order of magnitude, over a very wide range of $M$ [4,6,7,10–14].

Elaborating on the latter observation, Heglund proposed an empirical formula for estimating the mechanical power $P$ expended by an organism of mass $M$ in order to move itself at speed V [13]:

$$P = M \cdot (0.478 \cdot V^{1.53} + 0.685 \cdot V + 0.072) \tag{1}$$

Do similar regularities apply to artificial, human-made systems? If they do, how do they differ from those that apply to biological systems?.

We are aware of one observation reported in this regard: the speed of ships increases with mass to the power of 1/6 along the same curve as fishes, and planes similarly follow the curve for birds [15]. A potentially related observation was made by Marden and Allen [16] who found strong similarities in mass-force relations of animals and human-made motors. Isalgue's [15] observation is intriguing and raises further questions to ask: Does such a relation apply to ground-mobile systems? Does the relation include energy expenditures? What is the extent of differences between biological and artificial relations of power, speed, and mass? Our paper is the first to explore these questions and provide initial answers.

Our motivation in asking these questions combines foundational and applied interests. First, we seek fundamental insights into underlying mechanisms and limits, as well as opportunities to optimize the energy envelope [8] of future artificial systems. Second, with growing interest in bio-inspired approaches to robotics, we seek to learn about the tradeoffs in speed, mass, and power of biological systems, including their behavioral and terrain navigation tradeoffs [8], as these tradeoffs may apply to robotic system design. For example, the CoT for even successful legged robots like BigDog and ASIMO is much higher than for animals [17]. Advanced and challenging design approaches are required to achieve a CoT comparable to that of animals, e.g., the case of MIT's Cheetah robot [17,18].

If we were to know whether and to what extent the power-speed-mass relations of human-made systems exhibit a consistent relation to those of biological organisms, the designers of robotic systems would have additional guidance for specifying achievable performance targets. We illustrate this applied aspect later in the paper.

This study makes the following contributions. First, we find that artificial ground-mobile systems of multiple types and over a great range of scale comply with the same power-mass-speed relation–the Heglund formula (Eq 1)–proposed nearly 40 years ago for biological organisms [7]. To our knowledge, this has never before been reported. Second, we show that the

Heglund formula does not capture well the power-mass-speed relation for artificial ground-mobile systems at the extreme end of the mass scale, starting at approximately 35,000 kg and beyond. Third, we develop a modified formula in the spirit of Heglund's pioneering proposal but that captures the behavior of both biological and artificial systems over a greater range of mass–more than 11 orders of magnitude.

The paper is organized as follows. In the next section, we review the data we used in this study. The Results section describes our observations regarding the quantitative regularities we found in the data. After that, the Discussion section explores the meaning, implications, and limitations of our findings. The next section, An Application, offers an example of how our findings could apply to the development of future robotic vehicles. The paper ends with conclusions and recommendations for future work.

## Data

In this paper, we focus on expenditures of mechanical energy and on corresponding data. Mechanical energy expenditures should not be confused with those of metabolic energy [9] which considers consumption of chemical energy of the fuel (in vehicles) or food (in animals). For example, metabolic energy expenditure by an animal is typically derived from the rate of oxygen consumption, recorded during exercise, applying an energetic equivalent of 20.1 J to 1 ml of $O_2$ consumed, e.g., refs [19–21]. Metabolic energy is outside the scope of this paper.

In the case of vehicles, we take the mechanical power output of a vehicle's engine $P_e$ as approximately equal (for the purposes of our allometric study) to the mechanical power expended for the vehicle's locomotion. In doing so we neglect the energy that a vehicle may expend while stationary, e.g., for operating sensors or air conditioning, as well as the expenditures of mechanical energy for maintaining the operation of the engine itself, e.g., tanks' cooling fans that may take 10–15% of the engine output [22, p.258]. In steady state locomotion, at constant altitude, the entire mechanical power output of the engine $P_e$ is transformed eventually into heat and transferred to the environment of the vehicle (air and ground). This involves a complex chain of energy transformations ultimately ending in dissipative processes such as friction of sliding elements, gears, joints, and hysteresis in the rubber of tyres [22, p.228; 23].

Similarly, in the case of animals, we take the net mechanical power output of an animal's muscles $P_a$ as approximately equal the mechanical power expended for the animal's locomotion. In doing so we neglect the mechanical energy that an animal may expend while stationary. In steady state locomotion, at constant altitude, the entire mechanical power output of the muscles $P_a$ is transformed eventually into heat and transferred to the environment of the animal (air and ground). This involves a complex chain of energy transformations ultimately ending in dissipative, heat-generating processes such as inelastic deformations of tendons and muscles [24–26].

However, the conceptual similarity of $P_e$ and $P_a$ breaks down when we turn to the ways in which the mechanical power output is experimentally measured, and the corresponding availability of experimental data. In vehicles, the mechanical power output of an engine $P_e$ can be measured directly in several relatively simple ways, typically involving measuring the torque and angular velocity at the end of the engine's shaft, and for vehicles the data are usually available at least for the maximum "rated" power output (see [27]) approximately corresponding to the vehicle travelling at maximum rated speed and maximum load weight. In animals, on the other hand, measuring the rate at which muscles produce and output mechanical energy $P_a$ is extremely difficult [24] and the corresponding data are essentially unavailable.

Instead of $P_a$, experimental biologists measure so called external work of an animal, the rate of which we designate here as $P_{ext}$. This quantity in locomotion of an animal is derived using a

number of approaches [9]. Most of them assess the changes in the body's potential and kinetic energy. The movement of the body's center of mass can be determined either using a recording of the movement of a marker placed near the center of mass, or most commonly from the forces that the body exerts against the ground (or force plates), i.e., the ground reaction forces, often using the procedure popularized in [24]. Examples include refs [6,11–13].

For the purposes of our research, the challenge is that $P_{ext}$ is not directly comparable to $P_a$. One major difference between $P_{ext}$ and $P_a$ is that $P_{ext}$ includes the elastic energy stored in muscles and tendons of an animal [28] and as such can be significantly greater than $P_a$. We explore this matter in detail in the Discussion section, where we show that although $P_{ext}$ and $P_a$ are fundamentally different quantities, $P_{ext}$ can serve as an order-of-magnitude approximation of $P_a$.

In the next section of the paper, we use $P_{ext}$ as a surrogate of $P_a$. We show that $P_{ext}$ exhibits the same regularity, i.e., complies with the same formula, as $P_{e.}$ Then, in the Discussion section we return to deriving the relation between $P_{ext}$ and $P_a$, and show that $P_{ext}$ approximates $P_a$ well within an order of magnitude, appropriately for our allometric study.

For biological organisms, we obtained the data on speed, mass, and power $P_{ext}$ via a rigorous review of experimental literature, which we believe to be exhaustive, to the best of our knowledge. These included (see Supporting information) cockroaches, spiders, crabs, humans, quails, chipmunks, dogs, kangaroos, horses, kangaroo rats, ground squirrels, spring hares, wild turkeys, penguins, stump-tailed monkeys, lemurs, greater rheas, Asian elephants, sheep, frogs, and lizards.

For artificial ground-mobile systems, our data on speed, mass, and power $P_e$ represent diverse classes of defense-related systems (artillery systems, tanks, etc.) along with civilian transportation, construction, mining, and outdoor recreational vehicles (see Supporting information).

## Results

Being interested in the relation among speed, power, and mass of systems, we focus on how artificial and biological systems compare in terms of the previously described the Heglund formula, Eq (1). Remarkably, the data suggest that the Heglund formula, originally developed for animals of up to 70 kg of mass, applies rather well to artificial systems, e.g., vehicles at least 2–3 orders of magnitude heavier.

Fig 1 compares the actual system/organism power versus power predicted by the Heglund formula. (Here the term "actual" refers to experimentally measured or estimated values in the case of biological organisms, and experimental or design specification values in the case of artificial systems.) Visual inspection suggests that both biological and artificial systems generally follow the Heglund formula over a very wide range of masses and powers of systems.

There is, however, a visually notable deviation of actual power from the predicted power, approximately above 300–400 hp, where the data points refer mainly to tanks and heavy trucks.

Excluding, for the time being, all data points above the 35,000 kg limit, the R2 between predicted and actual power (on the log10 scale) for the entire set comprising both biological and artificial systems is 0.988. The R2 for biological organisms is 0.989, and R2 for artificial systems 0.945. To further assess whether the Heglund formula applies equally well to biological organisms and artificial systems below 35,000 kg, we performed an analysis of covariance (ANCOVA) to compare the slope and intercept of the regression line fitted to organism data to the slope and intercept of the regression line fitted to artificial system data (Fig 2). We

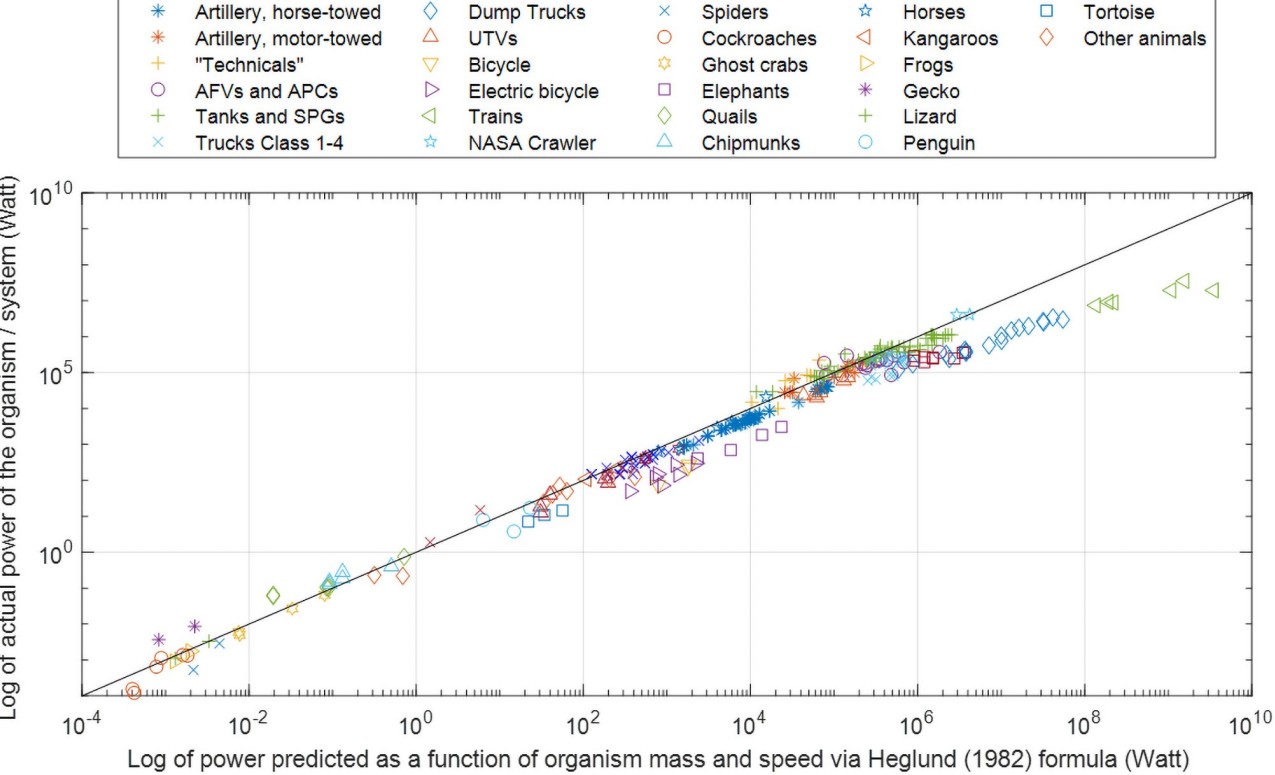

**Fig 1. Comparison of mechanical power predicted by the Heglund formula vs. the actual measured or specified power.** The diagonal line illustrates the close agreement of predicted vs. actual values. For organisms and systems with a weight of <35,000 kg, R2 is 0.98 (on the $\log_{10}$ basis; throughout the paper log refers to $\log_{10}$).

found no statistically significant difference between the intercepts (0.0160; $p$-value = 0.985) and the slopes (0.0214; $p$-value = 0.357).

To include systems with a mass greater than 35,000 kg, we developed a new formula, Eq (2). The formula takes inspiration from the Heglund formula, in the sense that it approximates the power expended by a system as a product of two functions–a function of the system's mass and a function of the system's speed. Unlike the Heglund formula, the proposed formula also takes into account the deviations at high mass values (Fig 3). Specifically, we assumed a piece-wise-linear model and used multiple linear regression to identify the coefficients in the following:

$$P = A \cdot M^b \cdot V^c, \tag{2}$$

or

$$log(P) = a + b \cdot log(M) + c \cdot log(V) \tag{2a}$$

where $a = 0.006$, $b = 0.986$, $c = 1.12$ for $M<35,000$kg, and otherwise, $a = 2.99$, $b = 0.489$, $c = 0.485$; here $P$ is power in watts, $M$ is mass in kilograms, and $V$ is speed in meters per second. The confidence intervals (at 0.95) for the intercepts and coefficients are listed in Table 1.

With this formula (which for the sake of brevity, we refer to as the KGP formula, from the initials of the authors), the R2 for the entire set of data (comprising both biological organisms and artificial systems) is 0.987 and the mean absolute percentage error (MAPE) is 0.0269 both

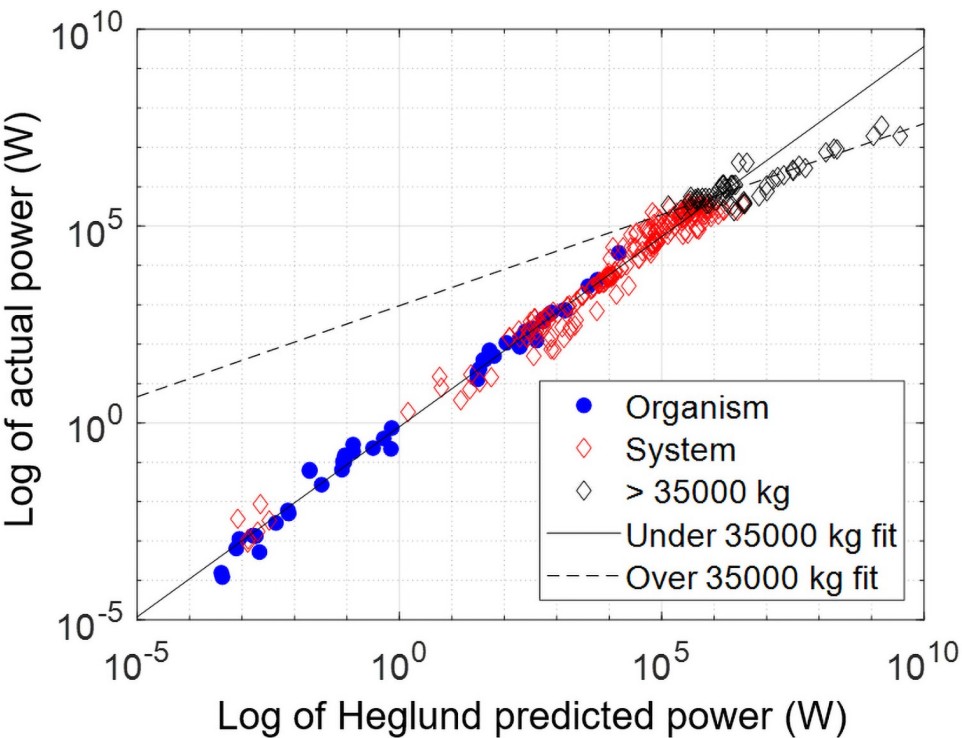

**Fig 2. Log of actual power versus predicted power for organisms and artificial systems.** Below 35,000 kg, both organisms and artificial systems fit well the line with slope 0.946 ± 0.125 and intercept -0.0830 ± 1.20. There is no statistically significant difference in slopes (p-value = 0.357) and intercepts (p-value = 0.985). For systems over 35,000 kg, however, the curve fit is distinctly different.

on the $\log_{10}$ basis. For $M<35,000$kg, a simplified formula $P = 1.01 \cdot M \cdot V$ provides a close approximation, with fit only marginally worse than by using Eq 2 and coefficients of Table 1.

As seen in Fig 3, the data for lower values of power (and, generally, lower mass) refer mainly to biological organisms, while the data for higher values of power (and mass) refer to artificial systems.

## Discussion

Let's begin the discussion by revisiting the question of relation between $P_a$ and $P_{ext}$ which we introduced earlier in the Data section. In the following, we build on the approach and data of Cavagna and co-workers [28]. Per Eqs 5 and 6 of [28], neglecting the negative work of muscles (as [28] does), the efficiency γ with which muscles transform chemical energy into positive work is $\gamma = P_a/P_{met} = (P_{ext} + P_{int}-P_{elast})/P_{met}$, where $P_a$ and $P_{ext}$ were introduced earlier, $P_{int}$ is the rate of work associated with movements of an animal's limbs with respect to its center of gravity, $P_{elast}$ is the power recycled via elastic storage within an animal's limbs, and $P_{met}$ is the chemical power input to muscles. Per [28], γ ranges from 0.2 to 0.3, and empirical values of $\alpha = P_{ext}/P_{met}$ range from 0.15 to 0.75, for a number of species (rhea, turkey, spring hare, kangaroo, dog and monkey).

Then, rearranging, the mechanical power output produced by muscles (comparable conceptually to the mechanical power output of an engine) $P_a = \gamma \cdot P_{met} = P_{ext} + P_{int}-P_{elast} = (\gamma/\alpha) \cdot P_{ext}$. Taking the range of values of γ and α mentioned above, we find that the values of $P_a/P_{ext} = \gamma/\alpha$ are in the range 0.27–2.0.

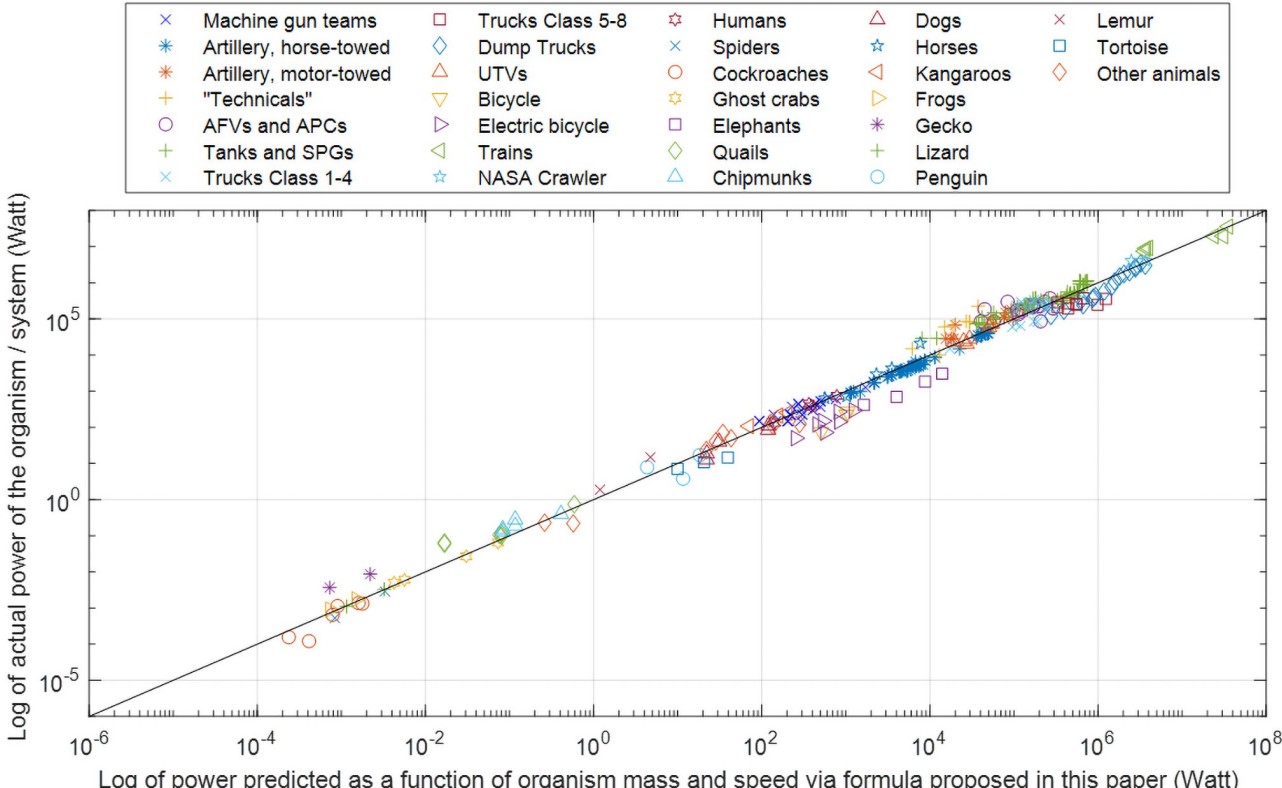

**Fig 3. Comparison of actual measured or specified power vs. mechanical power predicted by Eq (2).** The diagonal line illustrates the close agreement of predicted vs. actual values. For the entire set of organisms and systems, R2 is 0.987.

Therefore, although $P_{ext}$ and $P_a$ are not directly comparable, $P_{ext}$ can serve as an order-of-magnitude approximation of $P_a$. Although the range of values above suggest that $P_{ext}$ could be twice smaller or nearly 4 times larger than $P_a$, this is acceptable for our allometric, order-of-magnitude study. For example, to become an outlier in Fig 3 (see the discussion of outliers later in this section), a data point would have to disagree with the predicted value of power by a factor of approximately 5. And although the values of $\alpha$ provided in [28] cover only 6 species, Fig 3 illustrates that data for many other species fall into the same pattern, consistent with the premise that $P_{ext}$ can serve as an order-of-magnitude approximation of $P_a$.

With this, returning to the results of the preceding section, we see a strong similarity between biological and artificial systems in terms of the relation among speed, mass, and mechanical power of locomotion: both organisms and artificial systems agree closely with the Heglund formula (Fig 1) and its extension, the KGP formula (Fig 3 and Eq (2)).

This is remarkable for several reasons. First, the agreement covers an enormous range of mass and power values–over 11 orders of magnitude in mass, from a cockroach (less than 1 g)

**Table 1. Confidence intervals (at the 0.95 level) for intercepts and coefficients using Eq (2a).**

|  | N | a | b | c |
|---|---|---|---|---|
| Below 35,000 kg | 260 | [-0.0535, 0.0655] | [0.965, 1.01] | [1.05, 1.18] |
| Above 35,000 kg | 50 | [2.22, 3.75] | [0.315, 0.663] | [-0.239, 0.731] |

to a loaded freight train (nearly 100,000,000 kg) and 12 orders of magnitude in terms of mechanical power.

Second, the Heglund formula was originally based on data limited to only a few animals with masses not exceeding 70 kg. There is no obvious reason why it should continue to be as valid when extrapolated to systems of dramatically greater mass–up to 3 orders of magnitude greater, about 35,000 kg (Fig 1).

Third, the Heglund formula was developed originally for biological organisms only. It is notable that the formula is also valid for systems of entirely different morphology and functionality, such as trucks and tanks of up to 35,000 kg in mass.

Turning to the KGP formula, let us first mention outliers. Following [29], we determined data to be an outlier if the standardized residual was greater than three scaled median absolute deviation (MAD) from the median standardized residual. Out of 316 points in the data set, only 15 are outliers. Most of them are bicycles and elephants, not surprisingly as both of these have been discussed in prior literature as remarkably efficient in terms of energy cost of locomotion. Bicycles had been mentioned as outliers in [30] and elephants' external mechanical CoT had been found in [31] to be far lower than any other animal.

Continuing to explore the KGP formula, and particularly its coefficients (Table 1), we note that for organisms and systems below 35,000 kg, the exponent for mass $b$ is close to 1.0, similar to the Heglund formula (Eq 1). The exponent for speed $c$ is higher than 1.0, i.e., the power grows nonlinearly with speed, also similar to the Heglund formula. To put it differently, the mechanical CoT, $P/(M \cdot V)$, increases with speed. This is a trend common for many mobile systems, as reflected in the Gabrielli–von Karman diagram [32].

Then, as systems become heavier, approximately above 35,000 kg, the relation among power, mass, and speed enters a very different regime. As seen in Table 1, power depends on mass with an exponent significantly less than 1.0. Furthermore, the dependence of power on speed diminishes. To put it differently, the CoT (or the specific resistance in the Gabrielli–von Karman terminology) becomes less dependent on speed and diminishes with mass proportionately to $M^{(b-1)}$, where $b$ is between 0.315 and 0.663 (Table 1). What could explain this behavior?.

The literature offers numerous explanations for the diverse allometric relations among mass, body length, speed, and power expenditures of biological organisms. Typically, such explanations build on the organism's need to minimize energy expenditures or the need to maintain acceptable level of stress in bones (see Supporting information). In a similar spirit, we too offer an explanation of why, as we have shown in this paper, heavy mobile systems' CoT diminishes with mass, proportionately to $M^{(b-1)}$, where $b$ is between 0.315 and 0.663.

For a system like a heavy tank, a key constraint on its speed is the sheer stress $S$ it imposes on the ground, which cannot exceed a soil-dependent value of $S_{max}$ [33]. Such a system expends power $P \sim S \cdot F \cdot V$, where $F$ is the system's footprint–the area of contact with the ground [34]. A heavy system attempting to reach its maximum speed is likely to be limited by $S_{max}$, a constant for a given soil type and its moisture content [35]. Therefore, its power $P \sim S_{max} \cdot F \cdot V$. Then, $P/(M \cdot V) \sim (S_{max} \cdot F)/M$. Because the footprint is proportional to the square of a system's linear dimension, which in turn is proportional to $M^{1/3}$, we have $P/(M \cdot V) \sim (S_{max} \cdot M^{2/3})/M$, and therefore $P/(M \cdot V) \sim S_{max} \cdot M^{(-1/3)}$. This is qualitatively consistent with our findings.

Having offered an explanation for one of our key findings, we cannot however recommend giving it too much credence. We are reluctant to rely on explanations that are mono-casual in nature, e.g., based on a stress within the system or optimization of energy consumption, etc. To explore this point, let's consider the remarkable multiplicity of factors governing the maximum speed that a vehicle of a given gross weight can develop using a given source of power (e.g., an internal combustion engine).

For this purpose, we refer to the NATO Reference Mobility Model [35,36], extensively developed over several decades and experimentally validated on thousands of vehicle tests in multiple countries. For example, Vong et al. [37] explore how a given vehicle's speed can be limited by dozens of different constraints involving soil properties, terrain characteristics, drivers tolerance for shocks, vehicles geometry and features (see Supporting information).

The constraints that govern the maximum speed and power requirements of an artificial, engineered ground-mobile system are complex and multi-faceted, involving multiple heterogeneous factors and interacting among themselves in diverse ways. This may be even more so in case of intricate biological systems. As such, future work must look beyond mono-causal explanations for the allometric relations discussed in this paper.

## An application

The findings of this paper are particularly relevant to designing robotic, ground-mobile systems. For one thing, robotics tends to have a significant theoretical and practical affinity to bio-inspired approaches and analogies. The growing interest in legged robots is one example. As such, the common regularity of biological and artificial systems is of interest. Furthermore, like biological organisms, robotic systems used for defense applications are more likely to operate on offroad terrain than a typical transportation vehicle [38], and our data collection strategy includes a broad range of data relevant to offroad locomotion.

Determining the feasible yet ambitious targets for tradeoffs among the power, speed, and mass of future terrestrial robots, particularly for defense applications, is a difficult task. It is undesirable to base such targets on current experience, because military hardware is often developed and used for multiple years and even decades; therefore, the specifiers and designers of such hardware must base their targets–competitive yet achievable–on future technological opportunities not necessarily fully understood at the time of design.

To be sure, much research literature discusses detailed models for the analysis and design optimization of power-efficient robots, e.g., refs [17,18,34,39]. Here our intent is different: we explore how robotic applications can benefit from the KGP relation. Since the KGP formula allows the designer of a robotic system to estimate power requirements from the system's desired mass and speed, it may help serve as a preliminary check of the design's realism and potential performance bounds.

Here, we consider a preliminary design concept–a quadrupedal robotic "mule" that we call Exploratory Design for Mule-like Equipment Carrier (EDMEC). EDMEC weighs 600 kg and travels at a speed of 2 m/s. EDMEC is envisioned to carry a payload up to 90 kg [40] consisting of munitions, food, extra batteries, and communication equipment. This quadrupedal system will be able to travel in complex terrains (dense forests, rocky plains, and moderately difficult mountain trails) and over obstacles like building rubble. We estimated the total power required for EMDEC locomotion as 5 kW (see Supporting information).

However, the KGP formula predicts that a 600 kg vehicle moving at a speed of 2 m/s would need 1,200 W of power. This means that EMDEC consumes 4.1 times more power than predicted by the KGP formula. Is something wrong with EMDEC's design or our power consumption model? Not necessarily.

Consider Fig 4, where we plotted a number of robotics systems in comparison with the data in Fig 3. These systems include a number of commercially available or fielded systems (green asterisks), research systems (green squares), and the EMDEC concept (the purple star). Most of these data points are positioned well above the diagonal curve, indicating that the corresponding systems consume more power than the KGP formula predicts based on animals and conventional (tracked or wheeled) vehicles.

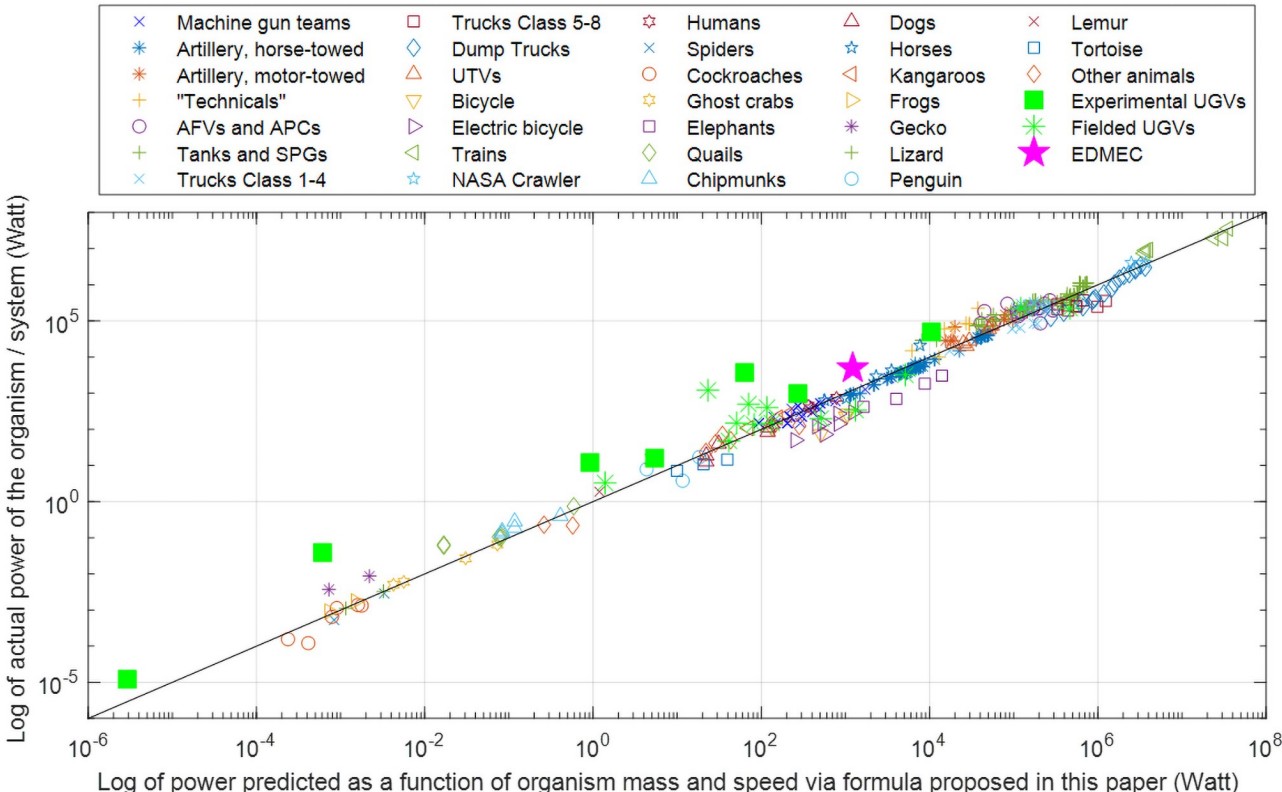

**Fig 4. Fielded and experimental robotic systems—Legged, tracked, and wheeled—Compared with the data in Fig 3.**

Particularly notable are the legged systems that lie significantly above the curve: the University of Maryland Micro-robot [41], NASA Valkyrie [42], and Boston Dynamics Atlas [43]. These systems consume 64, 52, and 60 times more power, respectively, than the curve predicts (Fig 4). Agility Robotics Digit [44] is another outlier. Two of the most efficient walking robots, the Cornell Ranger [38] and the MIT Cheetah [17], use 3.0 and 3.6 times the power predicted by the KGP curve, respectively.

On the other hand, we should also note there are three data points that are close to, or even below, the curve and whose power is of the same order of magnitude as EMDEC. These are wheeled or tracked systems (Clearpath Robotics Jackal [45] and Warthog [46], and iWarrior 710 [47]), which tend to be more efficient than their legged counterparts [17,48].

The relative inefficiency of legged systems is a common topic of discussions in literature, e.g., [17,49–51]. In fact, many ground vehicles lag behind their animal counterparts in movement efficiency when not moving on specially prepared surfaces like roads [15–17,52]. Much of the legged robots' inefficiency can be explained by the lack of passive elements like tendons in animals [4]. Passive elements allow for energy to be stored and recycled back into the system, thus increasing efficiency [17]. In fact, animal tendons can contribute up to 45% of the power required for locomotion in kangaroos and 40% in horses and camels [4]. Another reason for why artificial systems perform less efficiently than animals is due to difficulties and inefficiencies in sensing and path planning [53].

Although attractive for efficiency reasons, including passive elements in the design of a legged robotic system often comes at the cost of reducing the system's versatility. Passive elements that recycle energy back into the system restrict the movement of appendages and often need

to be empirically tuned to specific operating environments [17,54]. Without passive elements, a legged robot sacrifices some efficiency but gains adaptability to variable environments [55], which helps it move in difficult environments inaccessible to more efficient wheeled and tracked vehicles.

## Conclusions

We found that artificial ground-mobile systems–as diverse as ground robots, small utility vehicles, trucks, and tanks–exhibit a consistent regularity of relation among mass, power, and speed. For the range of mass from 1 g to 35,000 kg, this regularity is similar to the Heglund formula, known since 1980s and applied to a range of ground-mobile animals. Therefore, a single formula describes organisms and systems, from a cockroach to a battle tank.

These findings point to a fundamental similarity between biological and artificial locomotion that transcends great differences in morphology, mechanisms, materials, and behaviors.

We also show that for very heavy vehicles, ranging approximately 35,000–100,000,000 kg, the relation among power, mass, and speed enters a different regime. The CoT (or the specific resistance in the Gabrielli–von Karman terminology) becomes less dependent on speed and also diminishes with mass, proportionately to $M^{(b-1)}$, where b is on the order of 0.5.

To cover the two different regimes–both below and above 35,000 kg–we propose a piece-wise-linear regression formula that closely agrees with the available data. The agreement covers enormous ranges of mass and power values–over 11 orders of magnitude in mass, from a cockroach (less than 1 g) to a loaded freight train (nearly 100,000,000 kg), and over 12 orders of magnitude in terms of mechanical power.

When the proposed formula is considered in relation specifically to an important class of future ground vehicles–legged robots–we note a consistent inefficiency in current designs: the power consumption is from 3 to 60 times greater than predicted by the formula. In other words, today's legged robots are significantly less efficient than animals or tracked and wheeled vehicles of comparable speed and mass. This, however, may be the inevitable price to pay to allow them to adapt to diverse and difficult terrains.

With regard to theoretical explanations of these empirical findings, we note that the constraints governing the maximum speed and power requirements of an artificial ground-mobile system are numerous, complex, and multi-faceted, involving multiple heterogeneous factors and interacting among themselves in diverse ways. This may be even more so in case of intricate biological systems. As such, we find it difficult to give too much credence to theories that are mono-casual in nature, e.g., based on a stress within the system or optimization of energy consumption. Future work must explore the theoretical space beyond mono-causal explanations for the allometric relations discussed in this paper.

## Supporting information

**S1 File.**
(DOCX)

## Acknowledgments

The views presented in this paper are those of the authors and not of their employer. Phyllis Mcgovern of CCDC Army Research Laboratory (ARL) searched for multiple books and articles that supplied the raw data for this research. Jody Priddy of the U.S. Army Engineer Research and Development Center assisted with obtaining data related to numerous ground

vehicles. Carol Johnson, the ARL editor, improved the style and the grammar of the manuscript.

## Author Contributions

**Conceptualization:** Alexander Kott.

**Data curation:** Alexander Kott, Sean Gart, Jason Pusey.

**Formal analysis:** Alexander Kott, Sean Gart.

**Methodology:** Alexander Kott, Sean Gart, Jason Pusey.

**Visualization:** Alexander Kott, Sean Gart.

**Writing – original draft:** Alexander Kott, Sean Gart.

**Writing – review & editing:** Jason Pusey.

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
