## [Decision Letter · Decision Letter 0]

28 Aug 2020

PONE-D-20-21456

From Cockroaches to Tanks: the Same Power-Mass-Speed Relation Describes both Biological and Artificial Ground-Mobile Systems

PLOS ONE

Dear Dr. Kott,

Thank you for submitting your manuscript to PLOS ONE. After careful consideration, we feel that it has merit but does not fully meet PLOS ONE’s publication criteria as it currently stands. Therefore, we invite you to submit a revised version of the manuscript that addresses the points raised during the review process.

We look forward to receiving your revised manuscript.

Kind regards,

Hussain Md Abu Nyeem, Ph.D.

Academic Editor

PLOS ONE

Journal Requirements:

Reviewers' comments:

Reviewer's Responses to Questions

**Comments to the Author**

1. Is the manuscript technically sound, and do the data support the conclusions?

Reviewer #1: Partly

Reviewer #2: Yes

2. Has the statistical analysis been performed appropriately and rigorously? 

Reviewer #1: Yes

Reviewer #2: I Don't Know

3. Have the authors made all data underlying the findings in their manuscript fully available?

Reviewer #1: Yes

Reviewer #2: Yes

4. Is the manuscript presented in an intelligible fashion and written in standard English?

Reviewer #1: Yes

Reviewer #2: Yes

5. Review Comments to the Author

Reviewer #1: The paper investigates allometric relationship of mechanical power expended per unit of momentum (cost of transportation, CoT) for a wide range of natural and artificial systems exhibiting terrestrial locomotion. Authors find that the previously known relationship proposed for the mass range of 10^-3-10^1 kg by Heglund [13] holds over a much wider range of masses, 10^-3-10^4 kg, in animals and engineered systems alike. They also analyze the data available for engineered systems in the range of 10^4-10^6 kg and find that a qualitatively different scaling law applies for that range. Two linear functions describing CoT in the sub-ranges are fitted to the data. The resulting piecewise-linear relation holds for the whole range of masses that was considered (10^-3-10^6 kg). The relationship is additionally verified by checking that it holds on the data on sauropods and elephants that was not used in fitting of the empirical law, and finding that the data agrees with the law. Some existing ground-based robotic systems are considered in the context of the considered allometric relations and, for the most part, are found to be outliers. Implications of this fact for design of such systems are discussed.

All the claimed findings are, to the best of my knowledge, novel. While allometric studies that include both animals and engineered systems are not unprecedented (e.g. ref [39, 17] of the manuscript), no systematic studies of that sort were done before for terrestrial locomotion. Pre-existing work is properly cited.

Data collection methods used in the study are mostly sound. the only issue I found was the method in which the elephants and sauropods data was obtained. Mechanical power expended by the animals is derived from literature estimates of metabolic power by multiplying it by a constant factor. The factor is taken from the study on horses ([27], masses around 5e2 kg) and used for animals with masses at least two order of magnitude larger. Meanwhile, in Introduction the authors mention that the metabolic and mechanical cost of transportation scales differently. That makes the proportionate estimate dubious at best.

I ask that authors do one of the following:

1) provide a justification for proportional estimation of mechanical cost of transport from the metabolic cost, for the relevant mass range, or

2) provide a more careful estimation of mechanical cost of transport for elephants and sauropods and redo the related analysis and Figure 4, or

3) drop the elephant and sauropod data analysis from the paper altogether.

Another issue with this part of the study is that the data, while available in raw form in the primary sources, is processed to obtain the mechanical CoT estimates. It would be beneficial to make the processed data available.

Also, Figure 4 is cluttered. Please only show the relevant range of masses and consider dropping the data for animals and artificial systems other than elephants and sauropods.

This issue, however, is minor, as it the data is only used to provide additional support to a claim established with other data. Even if that part of the study is removed its primary claims can be sufficiently well supported.

Data analysis methods are sound. There is a minor presentation issue with ANCOVA analysis at the bottom of page 4: a single value is given as "difference" between slopes, and the value (0.964) is too large to be the true absolute difference between slopes that are close to 1. Please provide both of the slopes with confidence intervals.

Another minor issue encountered throughout the paper: values are reported to the third digit of the error margin. It is customary, at least in physical sciences, to round error margins down to one significant digit (two if the first one is 1) and also round the point estimate to the same decimal. For example, 0.964 \\pm 0.0224 becomes 0.96 \\pm 0.02. Similarly, for p-values usually only one significant digit is reported. Such conventions vary by the discipline, so I won't ask an adherence to this one, but in my opinion it could improve readability of the paper.

There is also a similar minor issue with the power estimate for EDMEC given in the Supplementary. The estimate is valid, but crude and must be treated as such. In light of that, the power consumption estimate of 4920W should be truncated to a single significant digit and presented as 5kW.

Language-wise, the manuscript is well-organized and accessible. There are a few places with typos and unclear language (listed below), but that is easily fixable and did not affect my ability to understand the intent of authors.

I recommend that the paper is accepted for publication after these issues are addressed.

----------------

Abstract:

"We show empirically not only..." - possible forgotten "that"

Introduction:

"metabolic rate scales approximately to the 3/4 power of the organism's mass" - awkward wording

In the discussion of CoTm, a typo: "P/(M*V)" -> "Pm/(M*V)"

Results:

p.10 last paragraph: "the slopes and intercepts of organism and system regression lines" - unclear

Supplementary:

The explanation of power expenditure section: "...is on the order of a magnitude or nearly constant magnitude of 10 body lengths per second..." - awkward

Reviewer #2: The authors test how predictable the relationship between mass and mechanical power is for biological and artificial systems based on a previously proposed formula. The authors extend the formula to consider systems of larger mass, where the previously proposed formula failed. The authors then use this metric to analyze the efficiency of current artificial legged walkers and find that for the most part they are several times more inefficient than predicted by the formula. They discuss the reasons for this departure and argue for more biologically inspired technologies as a direction forward. The manuscript is straightforward, well written, and technically sound.

6. PLOS authors have the option to publish the peer review history of their article (what does this mean?). If published, this will include your full peer review and any attached files.

Reviewer #1: **Yes: **Anton Bernatskiy

Reviewer #2: No

<gdiv></gdiv>

---

## [Author Response · Author response to Decision Letter 0]

29 Sep 2020

All responses are in the file submitted as part of the revised submission.

---

## [Decision Letter · Decision Letter 1]

9 Nov 2020

PONE-D-20-21456R1

From Cockroaches to Tanks: the Same Power-Mass-Speed Relation Describes both Biological and Artificial Ground-Mobile Systems

PLOS ONE

Dear Dr. Kott,

Thank you for submitting your manuscript to PLOS ONE. All the comments of the editor and earlier reviewers have been addressed well. However, a few questions about the comparison measures and the implications of the given data need further clarification. Therefore, we invite you to submit a revised version of the manuscript that addresses the points raised during the review process.

Note PLOS ONE strives to facilitate timely review with our continued effort to improve the speed and quality of the review process. However, please understand that due to unavoidable circumstances of the academic editor, your manuscript experienced an unusual delay this time. 

If applicable, we recommend that you deposit your laboratory protocols in protocols.io to enhance the reproducibility of your results. Protocols.io assigns your protocol its own identifier (DOI) so that it can be cited independently in the future. For instructions, see: http://journals.plos.org/plosone/s/submission-guidelines#loc-laboratory-protocols

We look forward to receiving your revised manuscript.

Kind regards,

Hussain Md Abu Nyeem, Ph.D.

Academic Editor

PLOS ONE

Reviewers' comments:

Reviewer's Responses to Questions

**Comments to the Author**

1. If the authors have adequately addressed your comments raised in a previous round of review and you feel that this manuscript is now acceptable for publication, you may indicate that here to bypass the “Comments to the Author” section, enter your conflict of interest statement in the “Confidential to Editor” section, and submit your "Accept" recommendation.

Reviewer #1: All comments have been addressed

Reviewer #3: (No Response)

2. Is the manuscript technically sound, and do the data support the conclusions?

Reviewer #1: Yes

Reviewer #3: Partly

3. Has the statistical analysis been performed appropriately and rigorously? 

Reviewer #1: Yes

Reviewer #3: Yes

4. Have the authors made all data underlying the findings in their manuscript fully available?

Reviewer #1: Yes

Reviewer #3: No

5. Is the manuscript presented in an intelligible fashion and written in standard English?

Reviewer #1: Yes

Reviewer #3: Yes

6. Review Comments to the Author

Reviewer #1: All of the issues I pointed out in my review have been fully addressed. The paper now satisfies all the requirements of PLOS ONE: it reports a novel result and it is technically sound. The reviewer thanks the authors for a good read!

Reviewer #3: The biggest challenges to this paper is in understanding the different metrics being used for ‘system/organism power’, perceiving whether these different metrics are reasonably comparable, and in interpreting the implications of the presented relationships.

Comparing metrics…

It is reasonable to begin with the Heglund (Cavagna, Taylor, Fedak) studies of the 1980s for the broad-brush scaling data and exponent fits. Of the three I recall (metabolic, center of mass and center of mass + ‘internal’) the third is selected for the initial scaling equation (Heglund et al. 1982 IV). This is the energy put into the center of mass added to the energies of the limbs about the center of mass (with this metric, it is assumed there is no transfer between the two). However, it is the second that is used for the majority of the biological data survey. Why not use center of mass power for both (Heglund et al., 1982 III)? Power proportional to mass and velocity (so a constant mechanical cost of transport): this appears to provide a better fit to the data too ( Table 1: a=0; b=1; c=1?). A bigger issue is that neither form for mechanical power is (and is acknowledged to be) a very poor predictor for the metabolic or ‘biological engine’ power.

But what is the ‘system’ power being described for the vehicles? It is presumably not the direct equivalent to the mechanical power demand of the animals – on flat level ground at constant speed this would be zero. Presumably it is the power supplied by the engine…? Under what condition? Maximum power or maximum speed?

And then, what of the horse-drawn guns? Is the work calculated the ‘animal’ way or the ‘machine’ way?

And so it is not immediately clear to me that the chosen animal and machine powers are reasonably comparable.

Implications…

If we believe there is some valid equivalency between the two metrics for power, the striking finding is the lack of evidence that a wheel improves matters. If I were to start off with a one-human-power runner (measured the animal way), and then put the human mass on a bicycle, I would hope one human-power to drive either a much more massive load or to travel much more quickly. That this is not evident across the data presented (compare 1000kg bull v.s truck) leads me to wonder whether 1) the two metrics for power are not comparable, or 2) trucks and tanks are horribly un-wheel-like. Where does a motorbike on a road sit on this line? And a small train on a railway track? It would be nice to see where a couple of ‘good’ (fast or economical) wheeled vehicles sit on this plot.

Minor points

To offer ‘data on request’ or ‘see pdf’ is not very 2020. The data are indeed available online

https://apps.dtic.mil/sti/pdfs/AD1098609.pdf

(Which may bring into question the issue of novelty – that is a policy decision outside my remit) but why not provide them as a spreadsheet so that the reader can easily start trying out their own statistical approaches?

The scaling relationship for the maximal speed in running animals falls over a bit at the larger sizes (Garland etc.)

Please double check the values taken from Heglund – there may be an issue with conversion to SI units. See Blickhan, R. & Full, R. J. 1987 Locomotion energetics of the ghost crab II. Mechanics of the center of mass during walking and running. J. Exp. Biol. 130, 155-174.

It appears odd to resort to using metabolics to derive the elephant ‘animal-power’. Does not Genin, Willems, Cavagna, Lair and Heglund (J. Exp. Biol., 2010) provide a more appropriate starting point? It may be that the elephant data point would then fit rather poorly: note that Genin et al. report CoM energy fluctuations 1/3rd that of smaller mammals.

And including sauropods as empirical points does feel a bit of a stretch.

7. PLOS authors have the option to publish the peer review history of their article (what does this mean?). If published, this will include your full peer review and any attached files.

If you choose “no”, your identity will remain anonymous, but your review may still be made public.

Reviewer #1: **Yes: **Anton Bernatskiy

Reviewer #3: No

<gdiv></gdiv>

---

## [Author Response · Author response to Decision Letter 1]

22 Dec 2020

The comments of the first 2 reviewers have been already answered to their satisfaction in previous revisions of the paper. This revision addresses the comments of Reviewer #3, and a detailed Response to Reviewer 3 has been provided as a file included in this submission.

---

## [Decision Letter · Decision Letter 2]

15 Jan 2021

PONE-D-20-21456R2

From Cockroaches to Tanks: the Same Power-Mass-Speed Relation Describes both Biological and Artificial Ground-Mobile Systems

PLOS ONE

Dear Dr. Kott,

Thank you for submitting your manuscript to PLOS ONE. After careful consideration, we feel that it has merit but does not fully meet PLOS ONE’s publication criteria as it currently stands. Therefore, we invite you to submit a revised version of the manuscript that addresses the points raised during the review process.

If applicable, we recommend that you deposit your laboratory protocols in protocols.io to enhance the reproducibility of your results. Protocols.io assigns your protocol its own identifier (DOI) so that it can be cited independently in the future. For instructions, see: http://journals.plos.org/plosone/s/submission-guidelines#loc-laboratory-protocols

We look forward to receiving your revised manuscript.

Kind regards,

Hussain Md Abu Nyeem, Ph.D.

Academic Editor

PLOS ONE

Additional Editor Comments (if provided):

The paper has been significantly improved and addressed all the major questions of the previous reviewers.

However, the academic editor is still interested in learning how the remaining questions of the current reviewer are addressed.

Reviewers' comments:

Reviewer's Responses to Questions

**Comments to the Author**

1. If the authors have adequately addressed your comments raised in a previous round of review and you feel that this manuscript is now acceptable for publication, you may indicate that here to bypass the “Comments to the Author” section, enter your conflict of interest statement in the “Confidential to Editor” section, and submit your "Accept" recommendation.

Reviewer #3: (No Response)

2. Is the manuscript technically sound, and do the data support the conclusions?

Reviewer #3: Partly

3. Has the statistical analysis been performed appropriately and rigorously? 

Reviewer #3: Yes

4. Have the authors made all data underlying the findings in their manuscript fully available?

Reviewer #3: Yes

5. Is the manuscript presented in an intelligible fashion and written in standard English?

Reviewer #3: Yes

6. Review Comments to the Author

Reviewer #3: The major issue with this paper is that it assumes comparability between two potentially quite different forms of power.

The vehicle values are for engine power, if possible, the ‘rated power’ (responses). This is the power (torque x angular velocity) at the end of the drive shaft. The rated power is useful in informing (once other losses are considered, and given suitable gearing) the capacity of a vehicle to accelerate, pull a load up an incline, or overcome drag.

The animal values are the rate of mechanical work of the center of mass during steady state locomotion.

Is it reasonable to assert an equivalency between ‘rated power’ and ‘center of mass power’? The difficulty is that the ‘center of mass power’ of a wheeled vehicle at any steady speed (even exceedingly high, and against drag) on level ground is zero. So ‘rated power’ cannot sensibly translate to ‘center of mass power’. But is the reverse likely? Does the center of mass power of an animal display something equivalent to its rated power? One can certainly imagine cases where this fails (consider cycling), but might it be reasonable for legged locomotion?

For the purposes of this paper, it would appear sufficient to state it as an assumption that it is reasonable, with whatever justification can be thought of and while acknowledging that others have also made this assumption.

At the moment, this is not sufficiently addressed. The term ‘rated power’ only appears in the responses – a description of the vehicle power close to the one I give above is required. Also in the responses is the justification:

“We do believe that for all systems (biological or artificial) in our study we are looking at fundamentally the same metrics: the amount of mechanical energy expended to propel the system.”

This is insufficient. Why is the center of mass power (that can be zero given wheels) ‘the amount of mechanical energy expended to propel the system’. Until an explicit work-around is given for this, any informed reader will view this study as a comparison of apples and oranges.

I wonder whether the biological literature is as exhaustive as claimed. I would suggest doing a citation search on Cavagna, 1975 ‘force platforms as ergometers’. Off the top of my head, are the reported values for cats, tortoise, penguins all unusable? Any lizards?

7. PLOS authors have the option to publish the peer review history of their article (what does this mean?). If published, this will include your full peer review and any attached files.

Reviewer #3: No

<gdiv></gdiv><gdiv></gdiv>

---

## [Author Response · Author response to Decision Letter 2]

21 Feb 2021

We accepted the reviewer's comments and provided revision accordingly. See the Response to Reviewers document provided with this submission.

---

## [Decision Letter · Decision Letter 3]

11 Mar 2021

From Cockroaches to Tanks: the Same Power-Mass-Speed Relation Describes both Biological and Artificial Ground-Mobile Systems

PONE-D-20-21456R3

Dear Dr. Kott,

We’re pleased to inform you that your manuscript has been judged scientifically suitable for publication and will be formally accepted for publication once it meets all outstanding technical requirements.

Kind regards,

Hussain Md Abu Nyeem, Ph.D.

Academic Editor

PLOS ONE

Additional Editor Comments:

The paper has been significantly improved and reasonably addressed the previous comments of the reviewers. Upon the 'ACCEPT' recommendations of the all the three reviewers, and seeing the improvements in the revised versions, the academic editor is now also convinced for its publication. 

Reviewers' comments:

Reviewer's Responses to Questions

**Comments to the Author**

1. If the authors have adequately addressed your comments raised in a previous round of review and you feel that this manuscript is now acceptable for publication, you may indicate that here to bypass the “Comments to the Author” section, enter your conflict of interest statement in the “Confidential to Editor” section, and submit your "Accept" recommendation.

Reviewer #3: All comments have been addressed

2. Is the manuscript technically sound, and do the data support the conclusions?

Reviewer #3: Yes

3. Has the statistical analysis been performed appropriately and rigorously? 

Reviewer #3: Yes

4. Have the authors made all data underlying the findings in their manuscript fully available?

Reviewer #3: Yes

5. Is the manuscript presented in an intelligible fashion and written in standard English?

Reviewer #3: Yes

6. Review Comments to the Author

Reviewer #3: I am grateful for the consideration given to my previous comments. I shall give a couple of additional thoughts, but certainly do not require further responses.

I would agree that the external mechanical power of a legged locomotor might be taken as a reasonable – at least order of magnitude – estimate of the minimum actuator work. Elastic mechanisms may play some role, and mean that it cannot be taken as an absolute minimum value, but the effect of these are likely – at the scales of interest here – to be negligible. However, the true actuator (muscle or engine) work may be much, much higher, and this may not necessarily be covered by an ‘order of magnitude’ argument. ‘Internal’ mechanical work demands may be significant; and external work could approximate zero. The Adaptive Suspension Vehicle (Waldron et al., 1984) carried its driver horizontally and steadily: external power approximately zero. However, the motor power is not.

So: cases can be imagined where the ‘apples with apples’ might not be true. How important this is (after all, horses really do not look at all like the ASV) to interpreting the current study can, in my view, now be left to the reader.

7. PLOS authors have the option to publish the peer review history of their article (what does this mean?). If published, this will include your full peer review and any attached files.

Reviewer #3: No

---

## [Editor Report · Acceptance letter]

15 Apr 2021

PONE-D-20-21456R3 

From cockroaches to tanks: the same power-mass-speed relation describes both biological and artificial ground-mobile systems 

Dear Dr. Kott:

I'm pleased to inform you that your manuscript has been deemed suitable for publication in PLOS ONE. Congratulations! Your manuscript is now with our production department. 

Kind regards, 

on behalf of

Dr. Hussain Md Abu Nyeem 

Academic Editor

PLOS ONE